# DeepOPF-GAF: Graph Self-Attention Framework for N-1 Security-Constrained Optimal Power Flow

## Abstract

N-1 Security-Constrained Optimal Power Flow (N-1 SCOPF) extends the conventional Optimal Power Flow (OPF) problem by ensuring secure and stable operation in all single-contingency scenarios. Solving OPF directly in large-scale power systems imposes a high computational burden, whereas compact approximation models, particularly multilayer perceptrons, have been introduced to improve efficiency. However, existing small-scale DNN-based models fail to meet the demands of highly dynamic grid topologies and multi-contingency solution requirements, due to their limited adaptability to topological changes and insufficient fitting capability. To bridge this gap, this paper proposes a graph self-attention-enhanced framework to optimize N-1 SCOPF solving. Specifically, a residual-based graph self-attention architecture is proposed to enable topological variation adaptation and scalable network expansion in depth and width. Furthermore, the Explained Variance Score (EVS) is introduced as a direct quantitative metric to evaluate the fitting performance of the proposed framework. Experimental results on the IEEE 9, 118, 300, and 2000-bus systems demonstrate that increasing the scale of the graph self-attention framework effectively enhances its fitting performance on N-1 SCOPF problems.

## 1 Introduction

The N-1 Security-Constrained Optimal Power Flow (N-1 SCOPF) problem is critical for the scheduling of practical power systems due to the N-1 security criterion. It ensures steady-state operation within specified safety and supply quality limits following N-1 contingencies, such as single line outages and generator failures(Alsac & Stott, 1974). Solving the N-1 SCOPF problem requires satisfying all load demands under each potential contingency scenario, which imposes a high computational burden, particularly in AC formulations. Inefficient N-1 SCOPF solving can hinder the flexible operation of power systems, especially at large scales. Therefore, it is important to explore efficient methods for solving the N-1 SCOPF problem.

Recently, machine learning methods have been studied to improve the speed of solving optimal power flow (OPF) problems. By analyzing and mapping large amounts of historical data, machine learning can derive solutions for specific scenarios. Studies have shown that deep learning-based approaches can even achieve speed improvements of two to three orders of magnitude compared to traditional solvers(Huang et al., 2021). To date, most OPF studies based on deep learning primarily use simple Multilayer Perceptrons (MLPs) for approximation(Huang et al., 2021; Pan et al., 2022; Nellikkath & Chatzivasileiadis, 2022; Dong et al., 2022; Zhou et al., 2022; Liu et al., 2022b; Velloso & Van Hentenryck, 2021; Pan et al., 2020). Among them, Huang et al. (2021) focus on high computational efficiency in solving the AC-OPF problem, where bus voltages are predicted, the remaining variables are reconstructed using power flow equations, a fast post-processing procedure is developed to enforce constraints, too. To further improve the feasibility of the obtained solutions, a penalty-based training method(Pan et al., 2022) is proposed along with a zero-order optimization technique(Agarwal et al., 2010) to compute gradients, enhancing computational efficiency and gradient stability. However, all these approaches struggle to handle scenarios with variable topologies due to the inherent limitations of MLPs in graph information aggregation.

For N-1 SCOPF scenario, Liu et al. (2022b) quantifies the probability of activating security constraints and employs a knowledge graph to record the system's operational states, aiming to improve the transferability of the learning model under different operating conditions. Nevertheless, this method relies on traditional iterative solvers, limiting its acceleration performance. There is also a strategy that directly predicts the achievable solutions for N-1 SCOPF combining Lagrangian duality(Velloso & Van Hentenryck, 2021). However, the approach is applied to the relatively simple DC-OPF, and during training, different constraints must be added for each fault iteratively. This complex operation may lead to instability in the training process.

Table 1: Summary of Nomenclature

| Notation | Definition |
|---|---|
| $V_i^{min}, V_i^{max}$ | minimum, maximum voltage magnitude of bus $i$ |
| $P_{gi}^{min}, P_{gi}^{max}$ | minimum, maximum real power of generator $i$ |
| $Q_{gi}^{min}, Q_{gi}^{max}$ | minimum, maximum reactive power of generator $i$ |
| $\theta_{ij}^{\min}, \theta_{ij}^{\max}$ | minimum, maximum voltage phase from bus $i$ to $j$ |
| $\mathcal{N}, \mathcal{N}_\mathcal{G}$ | sets of all buses, generation buses |
| $P_{ij}, Q_{ij}$ | active, reactive power flows from bus $i$ to $j$ |
| $g_{ij}, b_{ij}$ | conductance, susceptance from bus $i$ to $j$ |
| $S_{ij}^{\max}$ | branch flow limit from bus $i$ to $j$ |
| $V_i$ | voltage magnitude of bus $i$ |
| $P_{gi}$ | real power of generator $i$ |
| $Q_{gi}$ | reactive power of generator $i$ |
| $\theta_i$ | voltage phase of bus $i$ |
| $\theta_{ij}$ | given by $\theta_{ij} = \theta_i - \theta_j$, voltage phase from bus $i$ to $j$ |
| $\mathcal{E}$ | set of transmission lines |

With the advancement of research, OPF solutions that consider changeable topologies have become mainstream. A notable approach adopts branch admittance values for network topology representation, addressing the AC-OPF problem with flexible topology and line admittance(Zhou et al., 2022). However, this method faces significant limitations in practical applications, as it is difficult to modify nodes and add branches. The prime solution for addressing changeable topology in recent studies is to integrate with graph neural networks (GNNs)(Owerko et al., 2020; Liu et al., 2022a; Gao et al., 2023; Pham & Li, 2024). Among them, most are for standard OPF. To address the limitation in current research where node injection powers in output labels remain invariant to network topology changes, which severely impacts prediction performance, topology-related variables such as locational marginal prices and voltage magnitudes are introduced as outputs(Liu et al., 2022a). A physics-guided graph convolutional neural network for OPF is proposed(Gao et al., 2023), too. it presents an iterative feature construction method that encodes both physical characteristics and practical constraints into node vectors to consider the coupling and nonlinear characteristics of OPF. For N-1 SCOPF, the newest architecture called the Augmented Hierarchical Graph Neural Network is developed to predict key congested lines(Pham & Li, 2024). The approach only considers the key lines in the results and re-solves the simplified N-1 SCOPF problem. However, this method provides limited improvement in acceleration performance, and the simplified OPF model may overlook critical constraints present in the original N-1 SCOPF scenario. At the same time, other advanced networks have gradually been introduced into OPF solving(Yang et al., 2021; Dinh et al., 2021; Tran et al., 2024). However, the vast majority of current studies use small-scale MLPs or GNNs to achieve acceleration performance, while overlooking the inherent limitations in their fitting capabilities. Since N-1 SCOPF requires mapping all fault scenarios simultaneously, it further highlights the disadvantages of small networks in terms of fitting capability.

In light of these challenges, this paper proposes an advanced large-scale Graph Self-Attention Framework (GAF) to approximate solutions to the N-1 SCOPF problem, leveraging the enhanced fitting capacity of deeper and wider networks to achieve more feasible and accurate results. The main contributions are summarized as follows:

- To the best of our knowledge, this paper is the first to propose solving the N-1 SCOPF with larger-scale graph neural networks to enhance feasibility and precision.

- A framework based on Graph Self-Attention Networks(GATs) is developed to solve the N-1 SCOPF, considering both line and generator failures. By incorporating relevant parameters such as lines and generators, our approach can provide reasonable solutions for variable topologies.

- The Explained Variance Score (EVS) is applied as a central indicator of fitting quality, thereby extending traditional assessments of feasibility and real-time performance to include precision.

## 2 PRELIMINARIES

### 2.1 AC OPTIMAL POWER FLOW PROBLEM

Referring to(Huang et al., 2021), the standard AC-OPF problem can be formulated as follows:

$$\min \sum_{i \in \mathcal{N}_G} C_i \left( P_{gi} \right) \tag{1}$$

subject to:

$$P_{gi} - P_{di} = \sum_{j \in \mathcal{N}} V_i V_j \left( g_{ij} \cos \theta_{ij} + b_{ij} \sin \theta_{ij} \right), i \in \mathcal{N}, \tag{2}$$

$$Q_{gi} - Q_{di} = \sum_{j \in \mathcal{N}} V_i V_j \left( g_{ij} \sin \theta_{ij} - b_{ij} \cos \theta_{ij} \right), i \in \mathcal{N}, \tag{3}$$

$$P_{gi}^{\min} \leq P_{gi} \leq P_{gi}^{\max}, i \in \mathcal{N}_G, \tag{4}$$

$$Q_{gi}^{\min} \leq Q_{gi} \leq Q_{gi}^{\max}, i \in \mathcal{N}_G, \tag{5}$$

$$V_i^{\min} \leq V_i \leq V_i^{\max}, i \in \mathcal{N}, \tag{6}$$

$$\theta_{ij}^{\min} \leq \theta_{ij} \leq \theta_{ij}^{\max}, (i, j) \in \mathcal{E}, \tag{7}$$

$$P_{ij}^2 + Q_{ij}^2 \leq \left( S_{ij}^{\max} \right)^2, (i, j) \in \mathcal{E}, \tag{8}$$

The objective of the model is to minimize the cost of the generators, as shown in equation 1, subject to the constraints equation 2-equation 8. Among these, equation 2 and equation 3 represent the power flow constraints, equation 4 and equation 5 represent the active and reactive power constraints of the generators, equation 6 and equation 7 correspond to the voltage magnitude and phase angle constraints, and equation 8 represents the branch flow. The key notations are summarized in Table 1.

### 2.2 N-1 SCOPF PROBLEM

In this search, generator and transmission line faults are both considered. For generator in bus $j$ which is in malfunction, make $P_{gj}^{\min}$, $P_{gj}^{\max}$, $Q_{gj}^{\min}$ and $Q_{gj}^{\max}$ in equation 4 and equation 5 to zero, this means that equation 2-equation 4 simultaneously satisfy:

$$\begin{cases} P_{gj} = 0, \\ Q_{gj} = 0 \end{cases} \tag{9}$$

For faulted line $L_{ft}$, this paper assumes the connection between bus $f$ and bus $t$ is served, as like equation 9. equation 2,equation 3 ,equation 7 and equation 8 need to simultaneously satisfy:

$$\begin{cases} g_{ft}, g_{tf} = 0, \\ b_{ft}, b_{tf} = 0, \\ S_{ft}^{\max}, S_{tf}^{\max} = 0 \end{cases} \tag{10}$$

Totally, for generator faults, the N-1 SCOPF needs to satisfy equation 1-equation 9; for line faults, it needs to satisfy equation 1-equation 8 and equation 10.

## 3 METHOD

For N-1 SCOPF, compared to the standard OPF, the primary difference is that all N-1 contingency scenarios must be considered simultaneously. This implies that fitting the N-1 security-constrained scenarios at the same case level constitutes multi-task learning. To ensure high efficiency and robustness in this multi-task scenario, we draw upon the successful practices of large-scale language models in multi-task learning. Since the information handled by N-1 SCOPF is inherently graph-based, a graph self-attention mechanism that not only ensures excellent performance in fitting the optimization task but also preserves the model's adaptability to variable topological scenarios is introduced. Specifically, this paper employs an architecture based on stacked Graph Self-Attention and Graph Convolutional Networks to facilitate parallel inference across multiple tasks, thereby enhancing overall model performance.

### 3.1 GRAPH SELF-ATTENTION MECHANISM

The Graph Self-Attention Network(GAT) (Veličković et al., 2017) was proposed to leverage masked self-attention layers and stacking operations to assign distinct weights to different nodes within each node's neighborhood, thereby capturing local dependencies with greater precision. It has demonstrated superior performance in tasks, including node classification(Verma et al., 2023), link prediction(Yang et al., 2023), graph classification(Gao et al., 2021), etc. In this section, the construction layers of the Graph Self-Attention Network will be present.

For each layer, the node feature $\mathbf{h} = \left\{ \vec{h}_1, \vec{h}_2, \ldots, \vec{h}_N \right\}$, $\vec{h}_i \in \mathbb{R}^D$, where $N$ is the number of nodes, and $D$ is the dimension of each node feature. In the initial stage, a weight matrix $\mathbf{W}$, where $\mathbf{W} \in \mathbb{R}^{D' \times D}$, is applied to perform a parameterized linear transformation on the node feature $\mathbf{h}$, Subsequently, an attention learnable parameter vector $\mathbf{a}$, where $\mathbf{a} \in \mathbb{R}^{2D'}$, is applied, then apply **LeakyRelu** to get the attention coefficient. This can be formally expressed as:

$$e_{ij} = \text{LeakyReLU} \left( \mathbf{a}^\top [\mathbf{W}\mathbf{h}_i \| \mathbf{W}\mathbf{h}_j] \right), \tag{11}$$

where $e_{ij}$ indicates the importance of node $j$'s features to node $i$. The operator $\|$ represents the concatenation of vectors. To ensure that the coefficients are comparable across different nodes, the **softmax function** is further applied to normalize the attention coefficients on all choices of $j$, as follows:

$$\alpha_{ij} = \text{softmax}(e_{ij}) = \frac{\exp(e_{ij})}{\sum_{k \in \mathcal{N}(i)} \exp(e_{ik})}, \tag{12}$$

where $\mathcal{N}_i$ is the neighborhoods of node $i$ in the graph. Once the normalized attention coefficients are obtained, the corresponding linear combination of features can be computed, serving as the final output features for each node, as follows:

$$\mathbf{h}'_i = \sigma \left( \sum_{j \in \mathcal{N}_i} \alpha_{ij} \mathbf{W}\mathbf{h}_j \right), \tag{13}$$

Then, a multi-head attention mechanism is introduced, which employs multiple learnable parameter vectors $\mathbf{a}$ to capture richer information. The results are computed as follows:

$$\mathbf{h}'_i = \Big\|_{k=1}^{K} \sigma \left( \sum_{j \in \mathcal{N}_i} \alpha_{ij}^k \mathbf{W}^k \mathbf{h}_j \right), \tag{14}$$

Finally, the mean aggregation method is applied to obtain the final output:

$$\mathbf{h}'_i = \frac{1}{K} \sum_{k=1}^{K} \sigma \left( \sum_{j \in \mathcal{N}_i} \alpha_{ij}^k \mathbf{W}^k \mathbf{h}_j \right), \tag{15}$$

In order to comprehensively consider the variability of topologies and the stability of the model, this research constructs the architecture by stacking the aforementioned graph self-attention, graph convolutional, and linear layers.

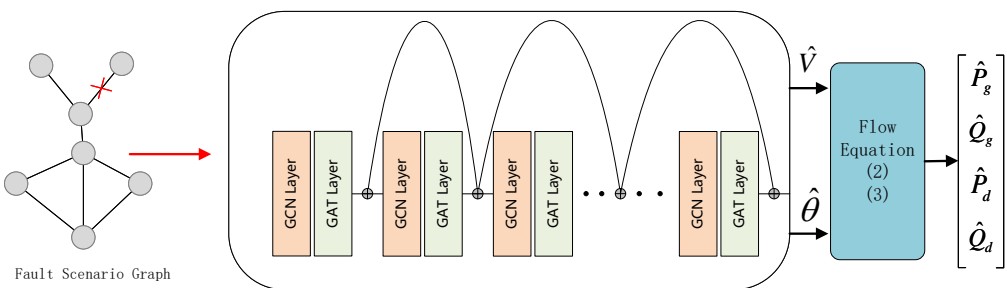

Figure 1: Schematic of proposed DeepOPF-GAF

## 3.2 THE PROPOSED GAF FOR N-1 SECURITY SCENARIO

Figure 1 illustrates the proposed regression architecture for the N-1 security scenario. For network design, graph self-attention layers and graph convolutional layers are integrated into blocks and stack. Meanwhile, residual mechanisms between the blocks are incorporated to mitigate issues such as vanishing gradients associated with deep networks. The input to the framework is the topological structure of the bus along with its features, where the node feature matrix encompasses information including $P_d$, $Q_d$, $P_g^{\max}$, $P_g^{\min}$. and the edge features represented by the adjacency matrix of bus system, These ensure that the architecture is capable of addressing a variety of fault scenarios simultaneously. For the output, the paper follows the approach outlined in (Huang et al., 2021), predicting only the voltage magnitude and angle information at each instance. Subsequently, the remaining solution variables, $P_g$ and $Q_g$, along with some auxiliary variables ($P_d$, $Q_d$) are directly computed through equation 2 and equation 3, bypassing the need to solve the nonlinear power flow equations. This also reduces the predictive burden on the network. Specifically, the loss function during the training process is as follows:

$$\mathcal{L}_{\mathcal{V}} = \sum_{i \in \mathcal{N}} \left\| \hat{V}_i - V_i \right\|_2^2, \quad \mathcal{L}_\theta = \sum_{i \in \mathcal{N}} \left\| \hat{\theta}_i - \theta_i \right\|_2^2,$$

and the training loss can be obtained:

$$L_{total} = \mathcal{L}_{\mathcal{V}} + \mathcal{L}_\theta \tag{16}$$

As can be seen from equation 16, the objective is to effectively fit information voltage magnitude, voltage angle, and power generation across various scenarios, making the selection of validation metrics critically important.

Currently, there is no universally recognized standard for evaluating the performance of machine learning in solving OPF problems; the prevailing benchmarks are as follows:

1. **Speedup**: This metric measures the acceleration performance by taking the ratio of the average time required by MIPS to solve the OPF problem to the average time taken by machine learning algorithms, denoted as $\eta_{sp}$.

2. **Optimality Loss**: This metric measures the average relative deviation between the optimal objective value obtained by MIPS and that achieved by machine learning algorithms, denoted as $\eta_{opt}$.

3. **Constraint Satisfaction**: This metric measures the feasibility of the obtained solution, primarily by evaluating the satisfaction rate of the variables for the constraints, including active power generation($\eta_{p^g}$), reactive power generation($\eta_{q^g}$), voltage magnitude($\eta_v$), voltage angle($\eta_\theta$).

4. **Load Satisfaction**: It measures the average satisfaction rate of the predicted load concerning the actual load, denoted as $\eta_{p^d}$ and $\eta_{q^d}$ for active and reactive load.

However, regarding the **Metric** 1, since conventional machine learning approaches for solving OPF do not incorporate the generator cost $C_i$, a lower optimality loss does not necessarily indicate better

fitting performance, as it also depends on the relationships among the generator cost. And for the **Metric** 2, since the constraints typically define a feasible region, when the feasible region is broad, high feasibility does not necessarily indicate a good fit. Experiments result in Figure 2 reveals that even small-scale networks can achieve high feasibility while exhibiting poor fitting performance, it suggests that the constraint satisfaction rate alone cannot consistently reflect the approach's effectiveness, and this will be further discussed in subsection 4.2 . In light of these observations, the EVS between the obtained variables and the true values are introduced as an evaluation metric. EVS is one of the commonly used metrics in regression tasks, reflecting the model's performance by assessing the extent to which the model accounts for the variability in the data. It is computed as follows:

$$\text{EVS}(\mathbf{y}, \hat{\mathbf{y}}) = 1 - \frac{\sum_{i=1}^{n} (y_i - \hat{y}_i)^2}{\sum_{i=1}^{n} (y_i - \bar{y})^2} \tag{17}$$

where $\hat{\mathbf{y}}$ and $\mathbf{y}$ represent the predicted vector and the actual vector, respectively, and $\bar{y}$ denotes the mean of the actual variable. EVS is confined to the range [0, 1]; values approaching 1 signify superior predictive performance, while those near 0 indicate inferior performance.

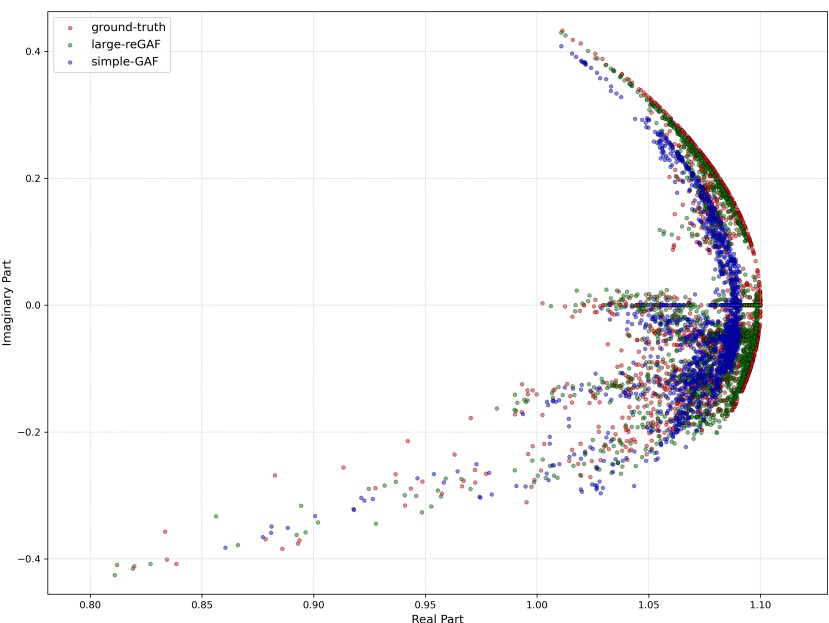

Figure 2: Scatter plot of predicted voltage data

# 4 NUMERICAL EXPERIMENTS

## 4.1 EXPERIMENT SETUP

The N-1 SCOPF simulations on power systems is conducted with varying load distributions (IEEE 9, 118, 300-bus system and a 2000-bus system) to obtain OPF solutions under different fault scenarios. During the training phase, the simulations are conducted utilizing four GeForce RTX 3090 GPUs, each with 24GB, Adam for optimization is employed with an initial learning rate of 1e-4 and the model is trained for 1000 epochs(for IEEE 2000-bus, it is 150 epochs). For the inference and testing stages, only a single GPU from this setup is utilized to ensure computational efficiency and consistency. Specifically, Matpower is utilized to simulate line outage and generator failure scenarios. For line outages, the faulty lines in the graph adjacency matrix is disconnected, and for generator failures, this paper invalidates the generator status at the corresponding nodes. Regarding the load information, load data is generated by uniformly sampling within the range of [70%,

130%] of the default values. Then, 100 samples per scenario are acquired, with an 80-20% split ratio for training and testing, respectively. Subsequently, the primal-dual interior-point method within the MATPOWER Interior-Point Solver(MIPS)(Zimmerman et al., 2010) is employed to obtain the ground truth. Since the constraints remain unchanged despite fault scenarios, it's hard to get solutions in certain scenarios like Islanding. For this, since the objective is to enhance the fitting capability of the model, unsolvable scenarios will be ignored. Specifically, for each fault scenario, it will attempt sampling up to 1000 times; if the number of successful samples does not reach 100, the scenario is deemed unsolvable.

For the models in experiment, simpleGAF, reGAF, and large-GAF, all adhere to the framework illustrated in Figure 1, with variations only in intermediate layer dimensions or network depth. Notably, simpleGAF excludes residual connections. The corresponding parameter sizes for each model are detailed in Table 5. For evaluation, including the metric EVS, the following metrics are employed to evaluate the performance of the deep-GAF framework: 1. **Speedup**, 2. **Optimality Loss**, 3. **Constraint Satisfaction**, 4. **Explained Variance Score**: It measures the model's fitting accuracy to the solution variables, encompassing voltage magnitude($\eta_v^{evs}$), voltage phase angle($\eta_\theta^{evs}$), active power generation($\eta_{p^g}^{evs}$), and reactive power generation($\eta_{q^g}^{evs}$).

Table 2: Comparision of performance for IEEE9-bus

| Metric | simpleGAF | reGAF | large-reGAF |
|---|---|---|---|
| $\eta_{\text{opt}}(\%)$ | 4.04 | -1.92 | **-1.26** |
| $\eta_V/\eta_\theta(\%)$ | – | – | – |
| $\eta_{p^g}/\eta_{q^g}(\%)$ | 99.89 / 99.95 | 99.94 / 100 | **99.98 / 100** |
| $\eta_v^{evs}/\eta_\theta^{evs}(\%)$ | 87.37 / 73.12 | 95.50 / 93.87 | **98.67 / 99.52** |
| $\eta_{p^g}^{evs}/\eta_{q^g}^{evs}(\%)$ | -74.56 / -111.84 | 90.46 / 61.44 | **79.90 / 95.20** |
| $\eta_{S_L}(\%)$ | 99.93 | 99.93 | 99.93 |
| $\eta_{p^d}/\eta_{q^d}(\%)$ | 87.38 / 82.15 | 93.89 / 90.77 | **97.08 / 93.35** |
| $\eta_{sp}$ | ×571 | ×465 | ×402 |

## 4.2 PERFORMANCE EVALUATION

The relationship between the feasibility of obtaining a solution and the performance of the fitting: The results in Table 2 demonstrate that for the 9-bus system, the feasibility indices($\eta_{p^g}$, $\eta_{q^g}$, $\eta_{S_L}$) of Simple-GAF, GAF, and Large-GAF are notably high and remarkably similar. However, their actual fit performance shows significant differences. Figure 2 presents the scatter plots of voltage values predicted by Simple-GAF and Large-reGAF against the true values. It can be observed that Simple-GAF exhibits significant deviations from the true values, this discrepancy cannot be reflected by the feasibility metrics alone. Therefore, we advocate for direct evaluation of fitting performance, which motivated the proposal of the EVS metric. As shown in Table 2, the EVS metric yields results that align well with the ground truth, demonstrating the validity of our proposed metric.

Table 3: Comparision of performance for IEEE118-bus

| Metric | simpleGAF | reGAF | large-reGAF |
|---|---|---|---|
| $\eta_{\text{opt}}(\%)$ | **-0.38** | 0.66 | -0.48 |
| $\eta_V/\eta_\theta(\%)$ | – | – | – |
| $\eta_{p^g}/\eta_{q^g}(\%)$ | **99.82** / 99.86 | 99.79 / 99.86 | 99.80 / **99.88** |
| $\eta_v^{evs}/\eta_\theta^{evs}(\%)$ | 98.55 / 98.89 | 99.46 / 99.33 | **99.73 / 99.87** |
| $\eta_{p^g}^{evs}/\eta_{q^g}^{evs}(\%)$ | 98.41 / 98.51 | 98.95 / 99.55 | **99.33 / 99.65** |
| $\eta_{S_L}(\%)$ | 100 | 100 | 100 |
| $\eta_{p^d}/\eta_{q^d}(\%)$ | 98.71 / 98.63 | 98.83 / 98.87 | **99.14 / 99.32** |
| $\eta_{sp}$ | ×533 | ×340 | ×121 |

The performance of models with different parameter sizes can be clearly observed from the results of 9-bus system in Table 2, where larger networks exhibit significantly superior performance over

Table 4: comparison of performance for IEEE300-bus and IEEE2000-bus

| Metric | IEEE 300-bus system | | 2000-bus system | |
| --- | --- | --- | --- | --- |
| | simpleGAF | large-reGAF | simpleGAF | large-reGAF |
| $\eta_{\text{opt}}(\%)$ | -0.63 | **0.03** | -1.43 | **0.76** |
| $\eta_V/\eta_\theta(\%)$ | - | - | - | - |
| $\eta_{p^g}/\eta_{q^g}(\%)$ | 99.89 / **99.86** | **99.91** / 99.82 | 99.31 / 99.65 | **99.52 / 99.85** |
| $\eta_v^{\text{evs}}/\eta_\theta^{\text{evs}}$ | 92.74 / 93.44 | **95.84 / 97.38** | 99.78 / 98.51 | **99.93 / 99.44** |
| $\eta_{p^g}^{\text{evs}}/\eta_{q^g}^{\text{evs}}$ | **97.98** / 97.23 | 96.34 / **98.33** | **99.60** / 90.39 | 98.28 / 98.25 |
| $\eta_{S_L}(\%)$ | >99.99 | >99.99 | >99.99 | >99.99 |
| $\eta_{p^d}/\eta_{q^d}(\%)$ | **98.81** / 97.40 | 98.15 / **98.08** | 98.12 / 96.25 | **98.62 / 98.27** |
| $\eta_{sp}$ | ×955 | ×125 | ×1712 | ×158 |

smaller ones on small-scale systems. Experiments are also conducted on IEEE 118/300/2000 systems for comparative analysis. From the results of 118-bus system, the networks at different size levels all demonstrate strong performance, and the fitting effect steadily improves as the network size increases in Table 3. Although larger networks may exhibit a slight decrease in the optimality (opt) metric, their overall performance is still superior because the electricity prices derived from the generated power outputs, which align more closely with the true solution variables, are more realistic. Table 4 demonstrates the effectiveness of our method on large-scale power systems (300 and 2000-bus systems). It can be concluded that adopting more complex and larger-parameter models helps mitigate the challenges of multi-scenario N-1 SCOPF, where solution patterns are often unclear.

Table 5: model parameter size

| Model | Parameters (M) |
| --- | --- |
| simpleGAF | 0.11 |
| reGAF | 7.02 |
| large-GAF(o) | 11.07 |
| large-reGAF | 11.06 |
| large-GCF | 11.05 |

In addition, all layers in Figure 1 are replaced with GAT layers to obtain large-reGAF(o), and then replace all layers with GCN layers to obtain large-reGCF, and conducted experiments comparing large-reGAF(o), large-reGCF, and the proposed architecture under the same parameter budget. As shown in Table 6, the proposed framework achieves state-of-the-art performance across most metrics, demonstrating its superior fitting capability for N-1 SCOPF problems. It can be explained this advantage primarily stems from the fact that in the N-1 SCOPF scenario, the topology changes are minimal, so GCN ensures stable learning, while GAT dynamically adjusts attention weights for the challenging-to-adapt portions. Therefore, combining both leads to superior performance.

Table 6: Comparision of performance for IEEE9-bus

| Metric | large-reGCF | large-reGAF(o) | large-reGAF |
| --- | --- | --- | --- |
| $\eta_{\text{opt}}(\%)$ | 7.25 | 11.68 | **-1.26** |
| $\eta_V/\eta_\theta(\%)$ | – | – | – |
| $\eta_{p^g}/\eta_{q^g}(\%)$ | 99.96 / 100 | 99.92 / 100 | **99.98 / 100** |
| $\eta_v^{evs}/\eta_\theta^{evs}(\%)$ | 97.98 / 99.03 | 97.93 / 99.22 | **98.67 / 99.52** |
| $\eta_{p^g}^{evs}/\eta_{q^g}^{evs}(\%)$ | 77.85 / **96.29** | 54.64 / 94.47 | **79.90** / 95.20 |
| $\eta_{S_L}(\%)$ | 99.93 | 99.93 | 99.93 |
| $\eta_{p^d}/\eta_{q^d}(\%)$ | 95.10 / **95.52** | 92.87 / 94.83 | **97.08** / 93.35 |
| $\eta_{sp}$ | ×448 | ×341 | ×402 |

## 5 CONCLUSION

A novel application of graph self-attention mechanisms is proposed to solve N-1 SCOPF problems. To address multiple contingency scenarios simultaneously, this study develops a residual-based graph self-attention framework and enhances its fitting capability through strategic expansion of network width. Simulation results demonstrate that the proposed architecture, with strong feature extraction capabilities, is both effective and essential for solving topology-dependent N-1 SCOPF problems. To evaluate the solution performance, this study introduces the EVS as a metric to assess the effectiveness of direct variable fitting, potentially providing valuable guidance for future training and testing methodologies in OPF-related tasks.

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
