# OpenReview forum: "DeepOPF-GAF: Graph Self-Attention Framework for N-1 Security-Constrained Optimal Power Flow"
_ICLR.cc/2026/Conference — Submitted to ICLR 2026_

### Official Review · Reviewer_qo6h · 2025-10-29

**Soundness:** 2
**Presentation:** 2
**Contribution:** 2
**Rating:** 2
**Confidence:** 4

**Summary:**

This paper introduces DeepOPF-GAF, a Graph Self-Attention Framework, for solving the N-1 Security-Constrained OPF (SCOPF) problem, which is at the center of power system optimization. This paper introduces a hybrid architecture combining GATs and GCNs. An additional evaluation metric called Explained Variance Score is introduced to assess fitting quality. Numerical results on different bus systems demonstrate the effectiveness of the proposed model, compared to other models with different sizes. However, there are technical issues with the N-1 SCOPF setup, model design choice, and baselines in evaluation.

**Strengths:**

- This paper aims to solve N-1 SCOPF problems, which is different from most of the existing studies on traditional OPF problems, given contingencies are important factors for operating a power system.
 - This paper proposes a regression architecture with graph self-attention and convolutional layers to predict voltage and phase angle with power flow handled by physical models, which looks sound.

**Weaknesses:**

- N-1 SCOPF often involves two stages: the first stage determines the optimal operating point for the normal so called base case, and the second stage ensures that this base case solution remains within operating limits after any single contingency has occurred. The setup of the N-1 SCOPF problem is unclear, and the evaluation section does not present the corresponding two-stage results.
- The claimed contribution about Explained Variance Score is not particularly significant, as it does not appear to be specifically tailored to the characteristics of N-1 SCOPF problems. Evaluation metrics for N-1 SCOPF would be more appreciated, given this work’s focus.
- The literature review mentions physics-guided approaches incorporating physical characteristics and practical constraints. It would be better to take such approaches as baselines for comparisons.
- In addition, the evaluation compares GAF variants only. But there are quite recent OPF studies including those cited and more that can be found in “ML OPF wiki” (hosted by ACM SIG Energy). Evaluation baselines can be enriched.
- The developed model chooses to take bus and line interactions as abstract dependencies instead of encoding power system physics, which may raise concerns about the model’s generalization under unseen circumstances. The justification of the design choice and possibly additional results would help address the concern.

**Questions:**

See the weaknesses.

---

### Official Review · Reviewer_PqGd · 2025-11-02

**Soundness:** 3
**Presentation:** 3
**Contribution:** 2
**Rating:** 4
**Confidence:** 4

**Summary:**

1.This paper addresses the N-1 Security-Constrained Optimal Power Flow (N-1 SCOPF) problem by proposing DeepOPF-GAF, a novel framework combining GCNs and GATs

2.The authors also introduce the Explained Variance Score (EVS) as a more robust metric to evaluate the model's fitting performance compared to traditional feasibility metrics.

3.The work demonstrates that the proposed method imporve overall accuracy across various IEEE test systems.

**Strengths:**

1. The work tries to address the critical N-1 SCOPF problem, which is highly important for ensuring the stability and security of modern power systems.

2. The paper is well-organized and clearly written, making the methodology and experimental setup easy to follow.

3.The introduction of the Explained Variance Score (EVS) for evaluating solution fidelity is a valuable contribution,

**Weaknesses:**

1.The core novelty of the proposed DeepOPF-GAF architecture is marginal. Simply combining GCN and GAT layers (which are structurally similar) in a stacked approach does not introduce significant theoretical insight for the OPF domain. The authors should better justify the specific design choice and its unique benefit for the minimal topology changes inherent in N-1 SCOPF.

2.While experiments cover different bus systems (up to 2000-bus), the overall dataset size remains small (only 4 systems). The paper would be strengthened by demonstrating the model's robustness and transferability across a wider variety of systems.

3.In ablation study, some experiments are missing: comparison between the general GNN and general GAT.

4.The figure does not clearly demonstrate the idea of this paper.

**Questions:**

please see the weaknesses

---

### Official Review · Reviewer_sBM2 · 2025-11-02

**Soundness:** 2
**Presentation:** 2
**Contribution:** 2
**Rating:** 2
**Confidence:** 4

**Summary:**

This paper proposes a Graph Self-Attention Framework (GAF) to solve N-1 Security-Constrained Optimal Power Flow (N-1 SCOPF) problems in power systems. The authors argue that existing deep learning approaches, particularly small-scale MLPs, have limited adaptability to topological changes and insufficient fitting capability for multi-contingency scenarios. They introduce a residual-based graph self-attention architecture combining GAT and GCN layers, and propose using Explained Variance Score (EVS) as a fitting performance metric. Experiments on IEEE 9, 118, 300, and 2000-bus systems demonstrate that larger networks improve fitting performance.

**Strengths:**

Application of the graph self-attention mechanism to electricity networks is interesting, it fits the existing literature on application of graph neural networks to electricity networks.

**Weaknesses:**

1) The novelty of application in the context of N-1 problem is rather overstated.

2) Missing contingency classification.

3) Scalability concerns inadequately addressed.

4) Limited architectural novelty.

5) Weak experimental design.

(For clarification see detailed comments and questions.)

**Questions:**

1) The novelty of application in the context of N-1 problem is rather overstated.

The N-1 consideration is relatively straightforward - simply setting certain parameters to zero (Equations 9-10) for each contingency scenario. The actual methodological contribution is applying larger graph neural networks, not fundamentally solving N-1 SCOPF in a novel way. Previous works (Velloso & Van Hentenryck, 2021; Liu et al., 2022b; Pham & Li, 2024) have already addressed N-1 scenarios, and other work discussed N-1 scenarios for larger grids using graph neural networks (Cambier van Nooten et al., 2025).

2) Missing contingency classification: In practical N-1 SCOPF, not all contingencies are equally important. The paper treats all contingencies uniformly without:
	⁃	predicting which line outages/generator failures will cause binding constraints,
	⁃	filtering or ranking contingencies by severity,
	⁃	any computational strategy to handle the combinatorial explosion in large systems (2000-bus system could have ~2000+ contingencies).

3) Scalability concerns inadequately addressed:
	⁃	The paper mentions ignoring "unsolvable scenarios" (lines 326-329) without discussing how many scenarios were discarded or the implications.
	⁃	For the 2000-bus system, how many total contingency scenarios were considered? The computational and memory requirements are not discussed
	⁃	Training with "100 samples per scenario" (line 331) seems limited for establishing generalization.

4) Limited architectural novelty: The proposed architecture is essentially a standard stacking of existing GAT and GCN layers with residual connections. The specific design choices (why alternate GAT/GCN layers? what depth/width trade-offs?) lack theoretical or empirical justification.

5) Weak experimental design:
	⁃	No comparison with recent GNN-based OPF methods (only comparing variants of their own architecture in Table 6)
	⁃	No comparison with the specifically cited N-1 SCOPF work (Pham & Li, 2024)
	⁃	No analysis of out-of-distribution performance or topology transfer

6) Generalization not addressed:
	⁃	All experiments use uniform sampling [70%, 130%] of base load - what about different load distributions?
	⁃	No testing on different network topologies than training
	⁃	No discussion of seasonal or daily load patterns

7) 	Edge Features Completely Unspecified. Line 244-246: "edge features represented by the adjacency matrix of bus system”. Critical missing information:
	⁃	What are the actual edge features? Just binary connectivity (0/1)?
	⁃	Or do edges include g_ij, b_ij (conductance/susceptance) from Equations 2-3?
	⁃	For line failures, Equation 10 sets g_ft = b_ft = 0, but nowhere is it explained how/where these are input to the model
	⁃	Equations 11-15 (GAT mechanism) show no edge features - only node features h_i, h_j
	⁃	Standard GAT uses only node features; if edge features exist, how are they incorporated?

8) Graph Modification Process Undefined. Lines 326-327: "faulty lines in the graph adjacency matrix is disconnected" and "invalidates the generator status”. Unclear:
	⁃	Is the adjacency edge removed entirely, or set to zero weight?
	⁃	Generator failures: P^max_g and Q^max_g are listed as inputs (line 244), but are they set to zero (per Eq. 9) or is there a separate binary mask feature?
	⁃	How does the model distinguish "no line exists" from "line exists but failed"?

9) No Topology Generalisation Experiments. Missing critical evaluations:
	⁃	Train on subset of contingencies, test on unseen contingencies
	⁃	Add/remove buses or lines not in training set
	⁃	Compare against fixed-topology baseline to quantify adaptation benefit

10) Comparison with Related Work Insufficient. Line 90-91 criticizes Zhou et al. (2022) as "difficult to modify nodes and add branches"
	⁃	No side-by-side comparison of topology handling mechanisms
	⁃	No experiments showing this method can add nodes/branches either
	⁃	Pham & Li (2024) specifically addresses N-1 SCOPF with GNNs
	⁃	See other references for relatable work.
11) Contradictory Claims. Line 107-108: "provide reasonable solutions for variable topologies" (emphasis on adaptation), but line 425-427: "topology changes are minimal" (downplays variation). Question: If N-1 changes are "minimal," why claim topology adaptation as a major contribution? A single line outage can dramatically alter power flows and constraint bindings.

---

### Official Review · Reviewer_ftJt · 2025-11-10

**Soundness:** 3
**Presentation:** 3
**Contribution:** 2
**Rating:** 2
**Confidence:** 5

**Summary:**

This paper proposes a graph attention-based neural network architecture for solving N-1 security-constrained optimal power flow (SCOPF) problems, where the power system topology is encoded as a graph. The model predicts voltage magnitudes and angles for multiple contingency scenarios in a unified framework.

**Strengths:**

1. The paper studies an interesting and practically relevant topic in power system optimization.
2. The use of a graph attention framework (GAF) is reasonable given the underlying network structure of the problem.
3. The presentation is clear and the paper is well-organized.

**Weaknesses:**

1. The contribution is limited; the work mainly involves applying an existing model architecture to a standard problem without substantial methodological innovation.
2. There is a lack of theoretical guarantees regarding the performance or optimality of the proposed approach.
3. It is unclear how the model adapts to different OPF formulations (e.g., AC vs. DC, cost functions, constraints), as the paper does not discuss this aspect.

**Questions:**

1. How does the proposed approach generalize to non-OPF problems, especially those where the graph structure is not inherent in the problem formulation?
2. The paper claims the model can handle various N-1 contingency scenarios, but how does it perform when the system topology changes more drastically (e.g., multiple simultaneous outages or reconfigurations)?
3. Can the authors clarify whether the model’s output remains physically feasible (e.g., respects all operational constraints) for unseen or rare contingency scenarios?
4. How sensitive is the model to the choice of input features or the quality of training data, especially when scaling up to larger or more complex networks?
5. The paper uses explained variance score (EVS) as a key metric; are there any domain-specific metrics (e.g., violation of operational limits, economic cost) where the model may perform poorly?
6.How does the approach compare to traditional optimization methods in terms of robustness and interpretability, especially for real-world deployment?
7. Is there any analysis on the computational complexity or scalability of the proposed architecture as system size increases?

---

### Meta-Review · Area_Chair_FYYR · 2026-01-05

**Summary:**

The paper proposes a method based on graph attention networks for solving N-1 Security-Constrained Optimal Power Flow problems. The authors also propose to use the Explained Variance Score (EVS) as a performance metric. The paper addresses an important problem and the application of GNNs is appropriate. The reviewers identified several critical limitations:
-  Lack of substantial methodological innovation. Reviewer sBM2 claims "The proposed architecture is essentially a standard stacking of existing GAT and GCN layers with residual connections." and that specific design choices lack theoretical or empirical justification. Similar concerns are echoed by other reviewers. The core contribution involves applying existing GNN architectures to a standard problem.
- Insufficient comparison with state-of-the-art GNN-based methods which also addressed the N-1 SCOPF problem (e.g. Pham & Li, 2024).
- Weak experimental validation that insufficiently probes the generalisation capabilities of the model to different load distributions, topologies and unseen contingencies.

Reviewer qo6h also raised a potential issue with the problem formulation: it is not clear whether the paper's setup and evaluation follow the typical two-stage approach. The reviewers also critiqued the lack of important implementation details including how topology changes are implemented and how edge features are handled.

The authors do no address any of the questions or issues raised. Three reviewers strongly recommend rejection (2). Reviewer PqGd has a slightly more positive score of 4 (marginally below the acceptance threshold) and appreciates the introduction of EVS as a metric. However, I rather agree with Reviewer qo6h which points out that EVS "does not appear to be specifically tailored to the characteristics of N-1 SCOPF problems.".

**Reviewer Concerns:**

No rebuttal.

**Reviewer Scores:**

No rebuttal.

---

### Decision · Program_Chairs · 2026-01-26

Reject